# Enkephalin-mediated modulation of basal somatic sensitivity by regulatory T cells in mice

Nicolas Aubert[1], Madeleine Purcarea[1], Julien Novarino[1], Julien Schopp[2], Alexis Audibert[3], Wangtianrui Li[1], Marie Fornier[1], Léonie Cagnet[1], Marie Naturel[1], Armanda Casrouge[1], Marie-Caroline Dieu-Nosjean[1], Nicolas Blanchard[3], Gilles Dietrich[4], Cedric Peirs[2], Gilles Marodon[1]*

[1]Centre d'Immunologie et des Maladies Infectieuses (CIMI-PARIS), INSERM, CNRS, Sorbonne Université, Paris, France; [2]Université Clermont Auvergne, CHU Clermont-Ferrand, INSERM, Neuro-Dol, Clermont Ferrand, France; [3]Toulouse Institute for Infectious and Inflammatory Diseases (Infinity), INSERM, CNRS, Université Toulouse III - Paul Sabatier (UPS), Toulouse, France; [4]Institut de Recherche sur la Santé Digestive (IRSD), Université de Toulouse, INSERM, INRAE, ENVT, Université Toulouse III - Paul Sabatier (UPS), Toulouse, France

*For correspondence:
gilles.marodon@inserm.fr

Competing interest: The authors declare that no competing interests exist.

**Abstract** CD4+CD25+Foxp3+ regulatory T cells (Treg) have been implicated in pain modulation in various inflammatory conditions. However, whether Treg cells hamper pain at steady state and by which mechanism is still unclear. From a meta-analysis of the transcriptomes of murine Treg and conventional T cells (Tconv), we observe that the proenkephalin gene (*Penk*), encoding the precursor of analgesic opioid peptides, ranks among the top 25 genes most enriched in Treg cells. We then present various evidence suggesting that *Penk* is regulated in part by members of the Tumor Necrosis Factor Receptor (TNFR) family and the transcription factor Basic leucine zipper transcription faatf-like (BATF). Using mice in which the promoter activity of *Penk* can be tracked with a fluorescent reporter, we also show that *Penk* expression is mostly detected in Treg and activated Tconv in non-inflammatory conditions in the colon and skin. Functionally, Treg cells proficient or deficient for *Penk* suppress equally well the proliferation of effector T cells in vitro and autoimmune colitis in vivo. In contrast, inducible ablation of *Penk* in Treg leads to heat hyperalgesia in both male and female mice. Overall, our results indicate that Treg might play a key role at modulating basal somatic sensitivity in mice through the production of analgesic opioid peptides.

## eLife assessment

This study presents a **valuable** finding on a new role of Foxp3+ regulatory T cells in sensory perception, which may have an impact on our understanding of somatosensory perception. The authors identified a previously unappreciated action of enkephalins released by immune cells in the resolution of pain and several upstream signals that can regulate the expression of the proenkephalin gene PENK in Foxp3+ Tregs. The generation of transgenic mice with conditional deletion of PENK in Foxp3+ cells and PENK fate-mapping is novel and generates **compelling** data; they also show a comprehensive analysis of Tregs in control and transgenic mice, longitudinal data on heat sensitivity and co-localization of PENK+ Tregs with thermal sensory neurons in the skin further supporting their hypothesis. The study would be of interest to the biologists working in the field of neuroimmunology and inflammation.

## Introduction

Regulatory T cells (Treg), characterized by the expression of the alpha chain of the interleukin-2 receptor CD25 and the transcription factor Foxp3 (*Hatzioannou et al., 2021*), are known to be key players in immunoregulation in both humans and mice; too few may lead to autoimmune diseases whereas too many may prevent an efficient immune response to cancer (*Kim et al., 2007*; *Nishikawa and Sakaguchi, 2010*).

Over the last few years, several new functions of Treg beyond immunoregulation have been identified in tissue regeneration or local regulation of metabolism (*Xiao et al., 2022*; *Meng et al., 2023*; *Shime et al., 2020*) for instance. In addition, there is accumulating evidence of a cross-talk between Treg and the nervous system. This includes promotion of oligodendrocyte differentiation or inhibition of neuroinflammation facilitating CNS repair process after brain injuries and preventing cognitive decline (*Dombrowski et al., 2017*; *Huang et al., 2020*; *Ito et al., 2019*; *Lemaitre et al., 2023*).

Furthermore, Treg have been involved in the regulation of pain in various models of nerve injury in rats and mice, such as in autoimmune neuritis or chronic constriction of the sciatic nerve (*Austin et al., 2012*; *Duffy et al., 2019*; *Kuhn et al., 2021*). Depletion of Treg has been associated with enhanced pain sensitivity whereas increased Treg number or activity limit pain hypersensitivity (*Lees et al., 2015*). Although the current view is that Treg controls pain through their immunosuppressive functions (*Bethea and Fischer, 2021*), whether Treg might regulate pain at steady state is currently unknown. Our results uncover a previously unknown function of Treg that modulates basal somatic sensitivity through the production of analgesic peptides derived from the proenkephalin *Penk* gene.

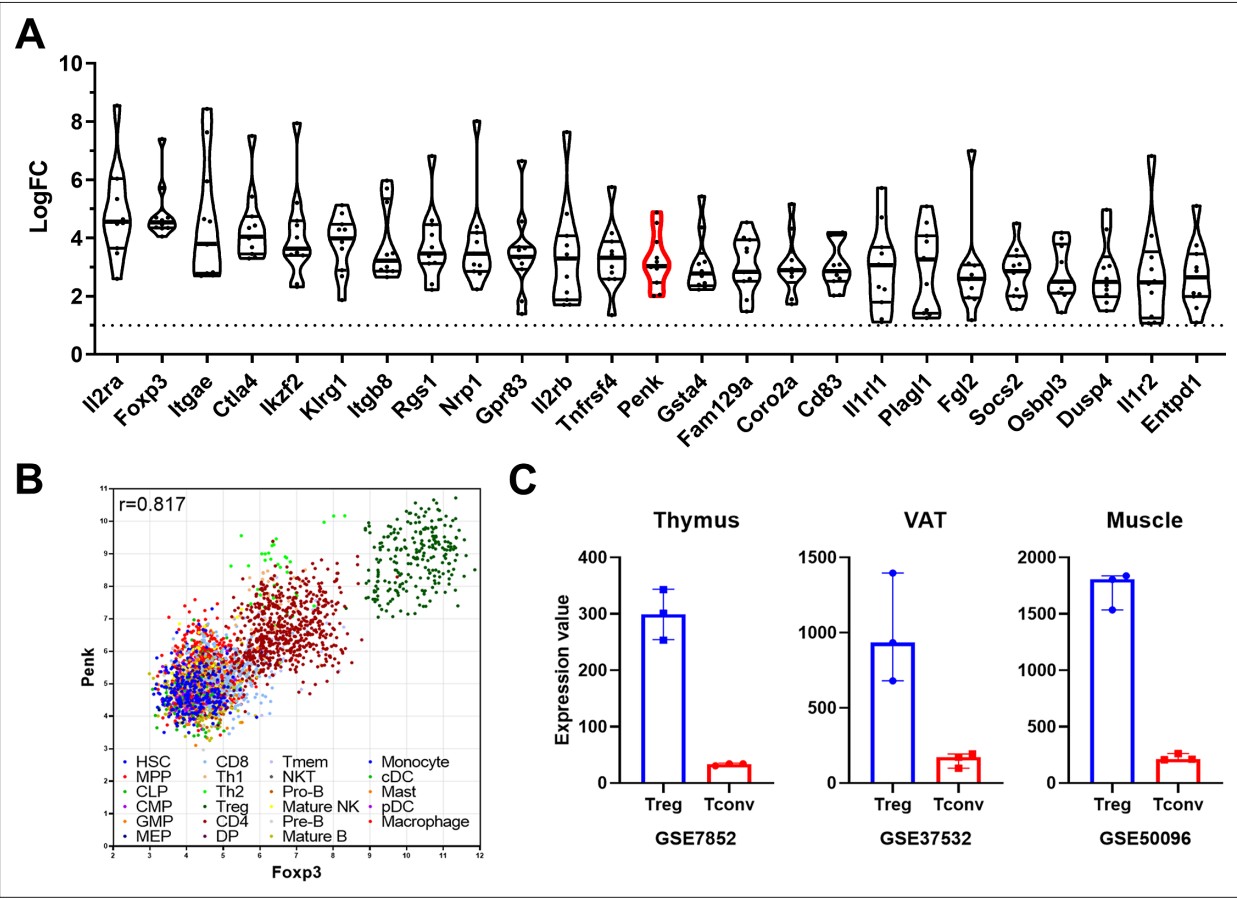

**Figure 1.** A meta-signature of murine regulatory T cell (Treg) and *Penk* mRNA expression in lymphoid and non-lymphoid organs. (**A**) Top 25 genes enriched in Treg compared to conventional T cell (Tconv) from lymphoid tissue from at least 10 of the 11 datasets analyzed ranked by fold change (LogFC) of mean expression relative to Tconv. The *Penk* gene is highlighted in red. (**B**) Correlation of *Penk* and *Foxp3* expression in all the cell types listed in the legend according to the Immuno-Navigator dataset. The Pearson correlation coefficient is indicated. Each dot is a sample, color coded as a subset according to the legend. (**C**) Expression of *Penk* in Treg (blue) and Tconv (red) isolated from thymus, visceral adipose tissue (VAT) and muscle. The source of the data is indicated below each graph.

**Table 1.** Characteristics of the datasets used for the regulatory T cell (Treg) meta-signature (NA = not available; LN = lymph nodes).

| Dataset | Genetic background | Age | Sex | Treg sorting | Tissue | Affymetrix Genome array |
|---|---|---|---|---|---|---|
| GSE103216 | C57Bl/6 | 6–8 weeks | Female | Foxp3-RFP | LN | 1.0 |
| GSE136582 | C57Bl/6 | 6–8 weeks | NA | Foxp3-eGFP | Spleen | 2.0 |
| GSE14308 | C57Bl/6 | NA | NA | CD25$^{high}$ | Spleen and LN | 2.0 |
| GSE15907 | C57Bl/6 | 6 weeks | Male | CD25$^{high}$ | Spleen | 1.0 |
| GSE17580 | C57Bl/6 | NA | Female | CD25$^{high}$ | Mesenteric LN | 2.0 |
| GSE24210 | C57Bl/6 | NA | NA | CD25$^{high}$ | Spleen and LN | 2.0 |
| GSE37532 | C57Bl/6 | 25 weeks | Male | CD25$^{high}$ | LN | 1.0 |
| GSE40685 | C57Bl/6 | NA | NA | Foxp3-GFP | Spleen and LN | 2.0 |
| GSE42021 | BALB/c | NA | NA | Foxp3-GFP | Spleen and LN | 2.0 |
| GSE50096 | C57Bl/6 | 6 weeks | NA | Foxp3-GFP | Spleen | 1.0 |
| GSE7460 | C57Bl/6 | 32–36 weeks | NA | CD25$^{high}$ | LN | 2.0 |

## Results

### A meta-signature of murine Treg

Using 11 available transcriptomes retrieved from the GEO website, we generated a Treg molecular signature at steady state in lymphoid tissues, significantly enhancing the robustness of our analysis compared to individual studies. The 25 most differentially expressed genes are depicted in *Figure 1A*, with the complete list provided in *Source data 1*. As expected from the sorting strategies used to isolate Treg (detailed in *Table 1* of the Materials and methods section), *Il2ra* and *Foxp3* emerged as the most differentially expressed genes in Treg compared to conventional T cell (Tconv). Several well-known Treg markers such as *CTLA-4*, *Itgae* (CD103), *Ikzf2* (Helios), *Tnfrsf4* (OX40), and *Tnfrsf9* (4-1BB) are also present on this list. Some genes, such as *Gpr83* and *Rgs1*, have been associated with Treg functions (*Flynn et al., 2023*; *Lu et al., 2007*) though their exact roles remain to be fully elucidated. Interestingly, the majority of genes on this list, including *Fam129a*, *Coro2a*, *Osbpl3*, and *Penk*, have unknown functions in Treg.

### Penk expression in Treg of lymphoid and non-lymphoid organs and tissues

Using the Immuno-Navigator database (*Vandenbon et al., 2016*), which provides a batch-corrected collection of RNA quantification across numerous studies, samples, and cell types, we confirm that *Penk* is highly correlated to *Foxp3* in lymphoid organs of mice ($r = 0.871$, *Figure 1B*). Notably, Treg samples exhibited the highest level of both *Foxp3* and *Penk*. While it has been previously reported that Th2 and Th1 cells can express *Penk* (*Boué et al., 2012*), they do so to a lesser extent than Treg (*Figure 1B*).

Additionally, using publicly available datasets comparing Treg and Tconv, we observe that *Penk* is enriched in the thymus, where Treg are generated, and is also present in peripheral tissues such as visceral adipose fat and muscles (*Figure 1C*). Therefore, the enrichment of *Penk* mRNA in Treg is intrinsic to their generation and is independent of their tissue localization at steady state.

### Penk mRNA expression is regulated by TNFR signaling and the BATF transcription factor

To explore possible mechanisms explaining the enrichment of *Penk* mRNA in Treg cells, we examined the Immuno-Navigator dataset to identify genes most correlated with *Penk* and with each other in Treg samples. We represent these correlations as a network where each node is a gene and each edge is a correlation above a certain threshold (*Figure 2A*). *Penk* expression is directly correlated

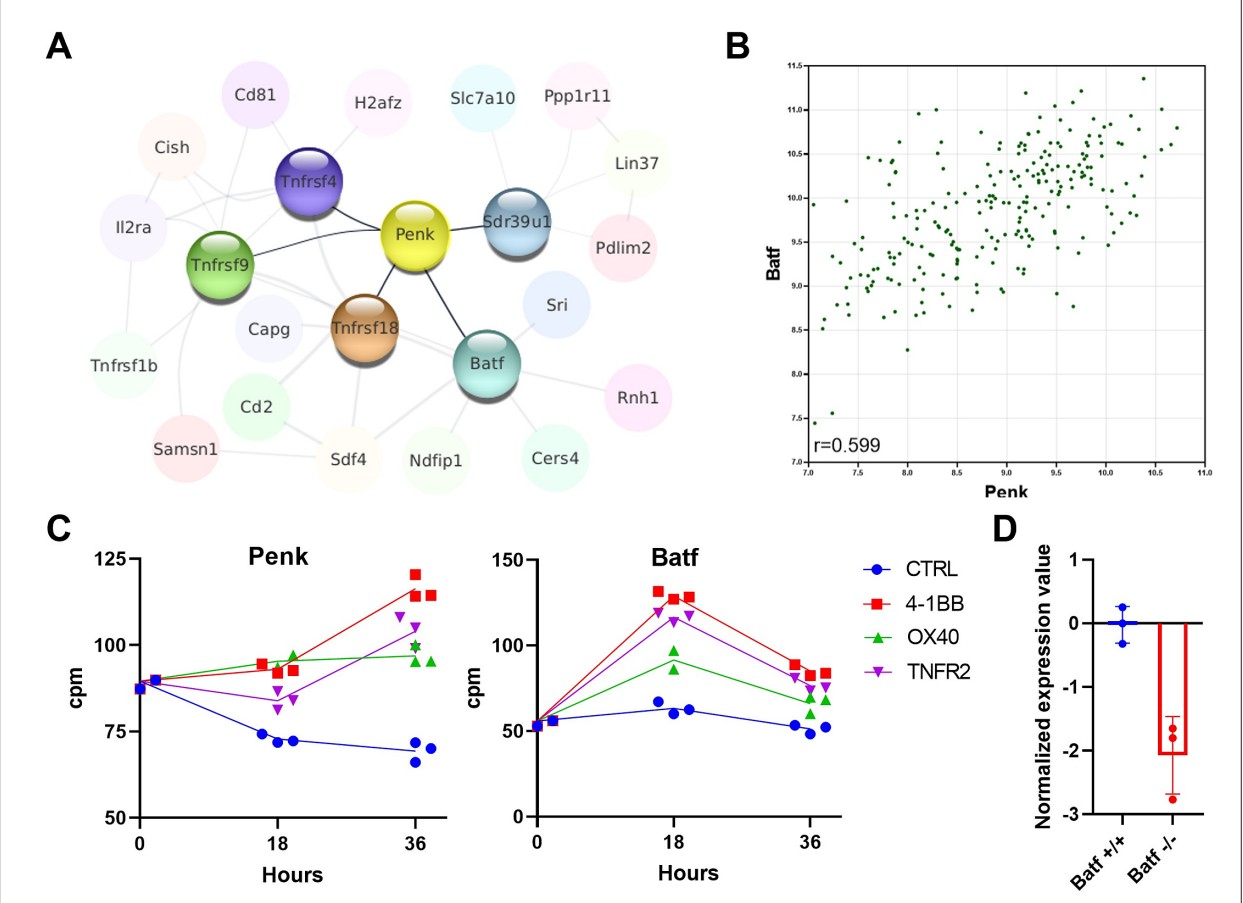

**Figure 2.** *Penk* expression is regulated by TNFR signaling and the BATF transcription factor. (**A**) A network of the genes most correlated to *Penk* in regulatory T cell (Treg) is shown. The Pearson correlation values were extracted from the Immuno-Navigator database (selecting only Treg in the analysis) and integrated into Cytoscape v3.7 (*Shannon et al., 2003*). Each node is a gene linked by edges with width proportional to the Pearson correlation (edge range: 0.538–0.758). (**B**) Illustration of the correlation between expression of *Penk* and *Batf* in Treg. Each dot is a sample from the Immuno-Navigator database. The Pearson correlation coefficient is indicated on the figure. (**C**) *Penk* and *Batf* mRNA expression after in vitro stimulation of purified Treg with the indicated TNFR agonists prior (0), and at 18 and 36 hr after stimulation. Each dot is a biological replicate from a single experiment. (**D**) GEO2R analysis of the GSE89656 dataset between wild-type control Treg (WT) and BATF-KO Treg.

The online version of this article includes the following figure supplement(s) for figure 2:

**Figure supplement 1.** Analysis of regulatory regions of the *Penk* gene.

to the expression of five genes: the TNF receptor family members, *Tnfrsf4* (OX40), *Tnfrsf9* (4-1BB), *Tnfrsf18* (GITR), the transcription factor *Batf*, and the short-chain dehydrogenase/reductase family 39U member 1 *Sdr39u1*. As an illustration, we show the correlation between *Batf* and *Penk* expression in Treg samples (*r* = 0.599, *Figure 2B*).

Furthermore, *Penk* is indirectly correlated with *Tnfrsf1b* (TNFR2), *Il2ra* (CD25), and *Cish*, a negative regulator of cytokine signaling. These strong correlations between several TNFR members, the transcription factor *Batf*, and *Penk* suggest a possible regulatory pathway. To explore this possibility, we reanalyzed our previously published dataset on the transcriptome of Treg stimulated with anti-CD3, anti-CD28 antibodies, and TNFR agonists in vitro (*Lubrano di Ricco et al., 2020*). We observe that addition of TNFR2, OX40, or 4-1BB agonists increases *Penk* expression at 36 hr post-stimulation relative to controls (*Figure 2C*). Interestingly, *Batf* is also increased with TNFR agonists but at an earlier time point (18 hr). Consistent with our hypothesis, a dramatic decrease in *Penk* expression is observed in Treg lacking *Batf* (*Hayatsu et al., 2017*; *Figure 2D*).

Furthermore, analysis of the UniBind database (*Puig et al., 2021*) revealed that the transcription factors Batf, Irf-4, Jun, and Fosl2 (AP-1 members), RelA (Nuclear Factor-kappa B signaling), and the

master Treg regulator Foxp3 have all been shown by ChIP-Seq to bind to the promoter/enhancer regions of *Penk* in various T cell subsets (*Figure 2—figure supplement 1*).

## Penk is predominantly expressed by Treg at steady state

Among other roles, *Batf* has been linked to tissue Treg differentiation in mice (*Burton et al., 2023*; *Delacher et al., 2020*; *Hayatsu et al., 2017*). Thus, we hypothesize that tissue Treg might be further enriched in *Penk* relative to lymphoid organs. To track *Penk*-expressing cells in vivo, we crossed a transgenic mouse model expressing Tamoxifen (TMX)-inducible Cre recombinase under the promoter of *Penk* (*Penk^Cre-ERT2*) with the *ROSA26^TdTomato* reporter mice. In these mice, any cell that expressed *Penk* at the time of TMX administration would become permanently tagged with the tdTomato reporter a few days later.

We investigated the expression of *Penk* mRNA in various immune cell types across multiple tissues by spectral flow cytometry, using a combination of lineage markers (*Supplementary file 1*) and an appropriate gating strategy (*Figure 3—figure supplement 1A*). To improve detection of *Penk* mRNA-expressing cells, we also used an anti-mCherry that cross-reacts with TdTomato (*Figure 3—figure supplement 1B*). In a Uniform Manifold Approximation and Projection (UMAP) representation of high-dimensional flow cytometry data, the projection of tdTomato expression (indicating *Penk* expression) aligns with the clusters of Treg and a small subset of activated CD4$^+$ T cells in the lymph nodes (LNs) (*Figure 3A*). Compared to LN, *Penk* expression encompasses entire clusters of Treg and activated T cells in the colon, likely due to the lower proportion of naïve T cells in this tissue (*Figure 3B*). Interestingly, in CD4$^+$ T cells (Tconv and Treg), *Penk* expression is higher in the activated CD62L$^{low}$ CD44$^{high}$ fraction compared to the naive CD62L$^{high}$ CD44$^{low}$ phenotype (*Figure 3—figure supplement 2*). As summarized in *Figure 3C*, the highest frequency of *Penk$^+$* cells is observed in Treg across all analyzed tissues, with the highest frequencies in the colon and the skin. *Penk* expression is also detected in Tconv of the colon and skin, but at lower frequencies than in Treg. All other cell types show low or undetectable *Penk* expression.

## Immunosuppressive functions of Treg are unaffected by the lack of enkephalins

To test whether the lack of enkephalins in Treg impacts their suppressive function, we generated mice deficient in enkephalins in the hematopoietic compartment by grafting bone marrow from *Penk* knock-out (KO) mice in immunodeficient Recombinase Activating Gene -2 RAG2-KO mice. As controls, RAG2-KO mice were grafted with bone marrow from wild-type (WT) littermates of KO mice. Several months after the graft, KO and WT Treg were sorted from lymphoid organs and tested in vitro for their ability to suppress the proliferation of Tconv. No significant difference in the suppression of Tconv proliferation is observed between WT and KO Treg (*Figure 4A*). Similarly, the addition of Naloxone, an irreversible blocker of enkephalin receptors, neither abolishes nor enhances the suppressive function of normal Treg cells (*Figure 4B*). Additionally, KO and WT Treg equally prevented the occurrence of autoimmune colitis induced by the transfer of naive Tconv cells into RAG2-KO mice (*Figure 4C*).

Overall, these results indicate that enkephalins are not major players in the suppressive functions of Treg cells both in vitro and in vivo.

## Heat hyperalgesia in mice deficient for Penk in Treg

To determine if enkephalins produced by Treg affect pain at steady state, we generated mice deficient in *Penk* by crossing TMX-inducible Cre recombinase under the control of the *Foxp3* promoter (*Foxp3^Cre-ERT2*) with mice transgenic for LoxP sequences flanking exon 2 of *Penk* (*Penk^flox*). In these mice (hereafter referred to as LOX), any cell expressing Foxp3 at the time of TMX administration would become deficient for exon 2 of *Penk* a few days later, hence lacking enkephalins.

Using flow cytometry, we observe that *Penk* mRNA expression is reduced by more than half in TMX-treated LOX mice compared to WT mice (*Figure 5—figure supplement 1*). Consistent with the *Penk*-Cre reporter mouse data, *Penk* mRNA expression is very low in non-Treg (CD4$^+$Foxp3$^-$ or CD8$^+$ cells) and does not differ significantly between LOX and WT mice. Thus, TMX treatment specifically reduces *Penk* mRNA expression in Treg cells of LOX mice.

Since exon 2 is the precursor of Met-Enkephalin, an endogenous opioid that affects thermal pain sensation (*Aman et al., 2016*), we evaluated the sensitivity of these mice to pain induced by heat

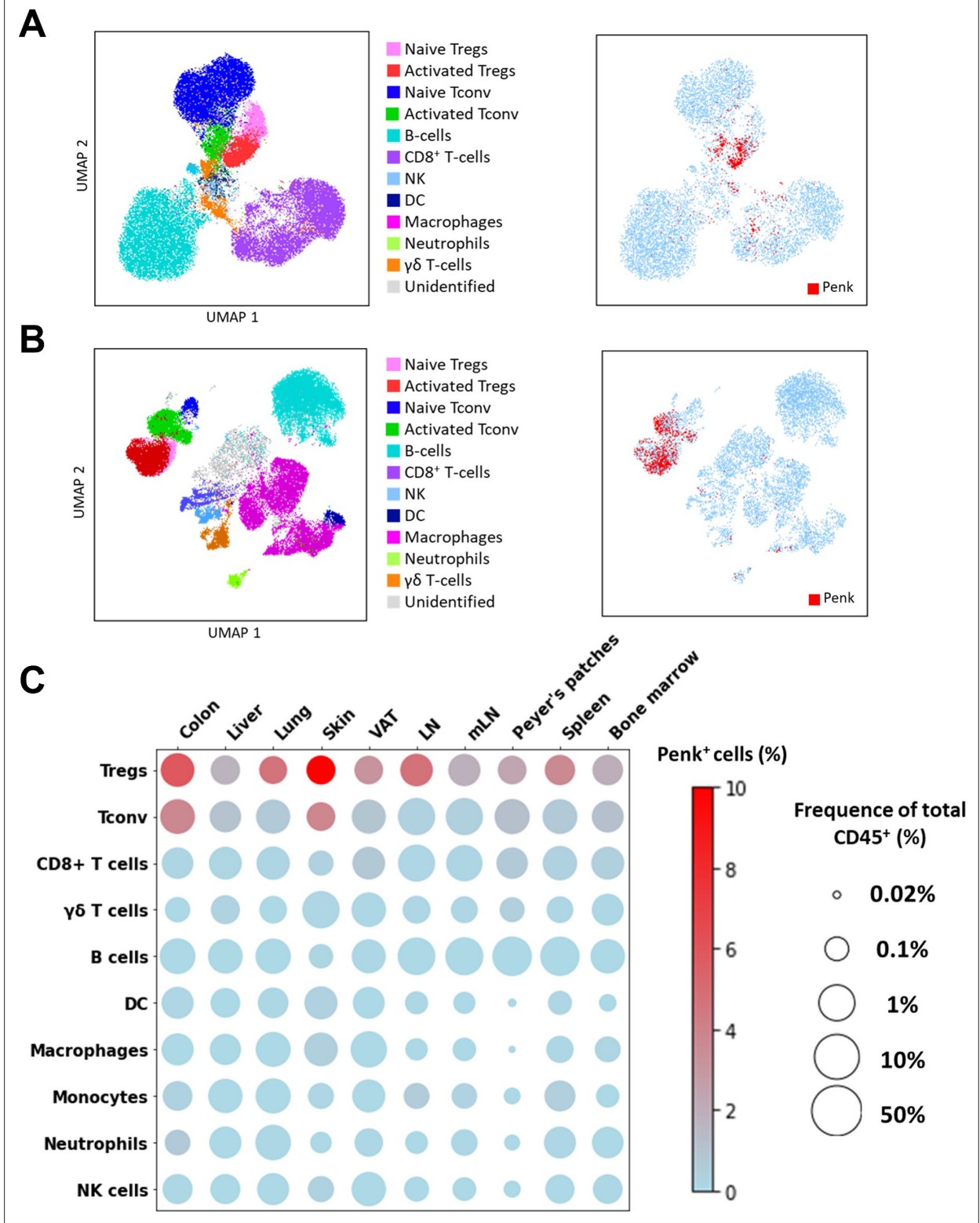

**Figure 3.** *Penk* is predominantly expressed by regulatory T cell (Treg) at steady state. (**A–B**) (Left) UMAP representing all major cell types indicated in the figures determined by flow cytometry from lymph nodes (**A**) and colon (**B**). (Right) Projection of tdTomato expression on the UMAP of lymph nodes and colon. Subsets were manually gated as depicted in Figure S1A. (**C**) Bubble plot displaying the average population size and frequencies of Penk+ cells for the listed cell populations and organs. Population size was calculated as the percentage out of total CD45+ single cells and represented on a log10

*Figure 3 continued on next page*

*Figure 3 continued*

scale (*n* = 3 mice for the visceral adipose tissue [VAT], spleen, and bone marrow, *n* = 6 mice for the other groups; results cumulative of two independent experiments).

The online version of this article includes the following figure supplement(s) for figure 3:

**Figure supplement 1.** Mapping *Penk* expression in multiple tissues and cell types.

**Figure supplement 2.** *Penk* expression in conventional T cell (Tconv) and regulatory T cell (Treg) according to activation status in lymph nodes.

(*Figure 5*). As controls, we used mice expressing the Cre recombinase and WT at the locus of LoxP sequences insertion. Mice were treated with TMX and evaluated for heat sensitivity at 10 different time points (four before and six after administration of TMX, 2–3 days apart). Under these conditions, a significant trend toward lower latency periods (indicative of heat hyperalgesia) is observed in the LOX group compared to WT mice (*Figure 5—figure supplement 2A*). Interestingly, the effect is not apparent until 7 days after TMX administration. Before TMX administration, WT and LOX mice do not differ in their response to heat. However, after TMX administration, LOX mice develop hyperalgesia compared to WT mice, with a 20% reduction in their median latency period from day 7 onwards (8.1 vs 6.6 s) (*Figure 5A*). The effect of *Penk* deletion in Treg on heat hyperalgesia is sex independent, as it is observed in both females and males (*Figure 5—figure supplement 2B, C*). Moreover, this thermal hyperalgesia in LOX mice is reproduced with a different test in an independent laboratory where WT and LOX mice were sent for further behavioral tests (*Figure 5—figure supplement 2D*). Indeed, tests for innocuous (Von Frey, cotton swab tests) and noxious (pin prick test) mechanical sensitivity, as well

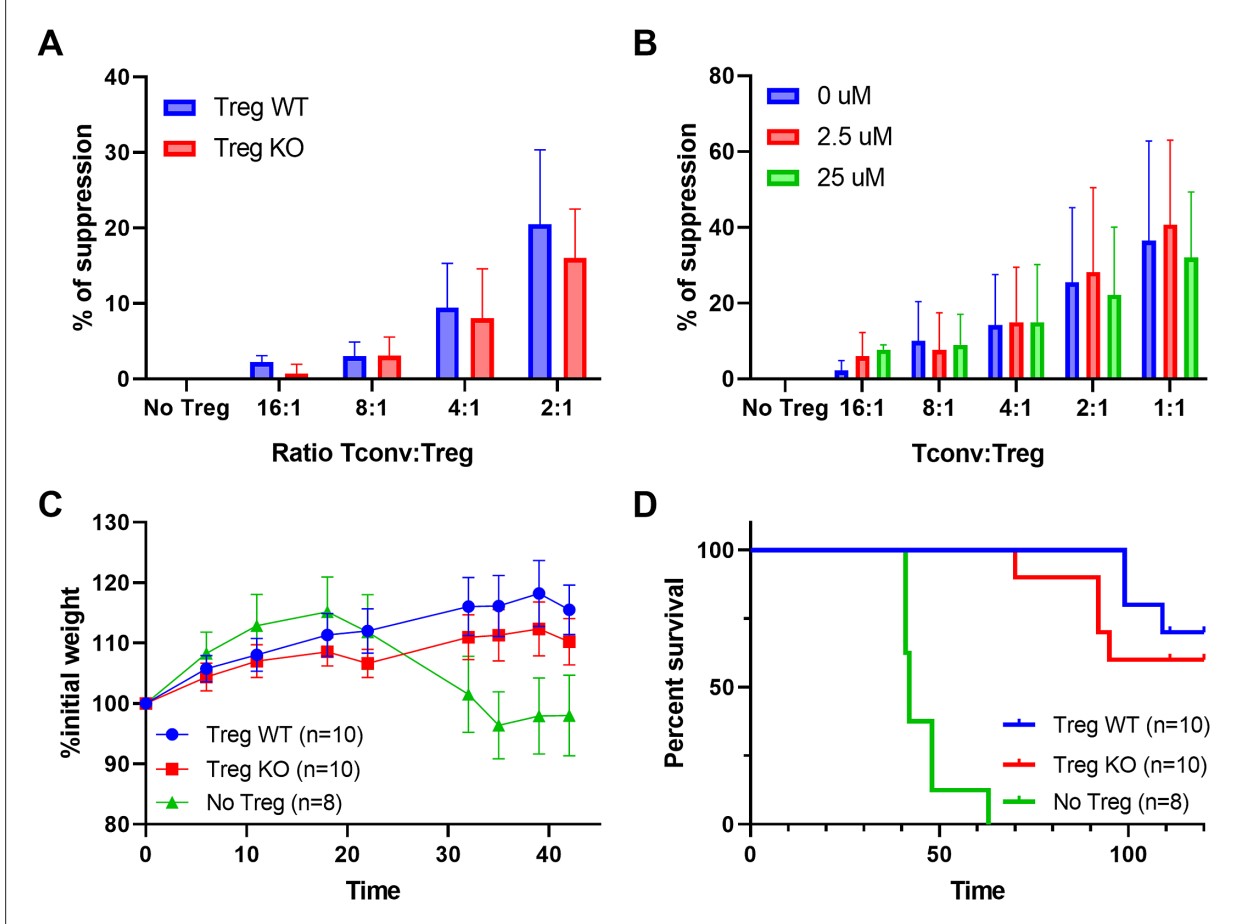

**Figure 4.** Immunosuppressive functions of regulatory T cell (Treg) are unaffected by the lack of enkephalins. (**A**) In vitro suppression of wild-type (WT) conventional T cell (Tconv) proliferation by WT or Penk knock-out (KO) Tregs. (**B**) In vitro suppression of WT Tconv proliferation by Penk WT Tregs in presence of Naloxone. (**C**) Body weight and (**D**) survival of Rag2$^{-/-}$ mice transferred with Tconv and Treg from Penk-WT or Penk-KO chimeric mice (2:1 ratio), as described in the methods. All data are cumulative of two independent experiments.

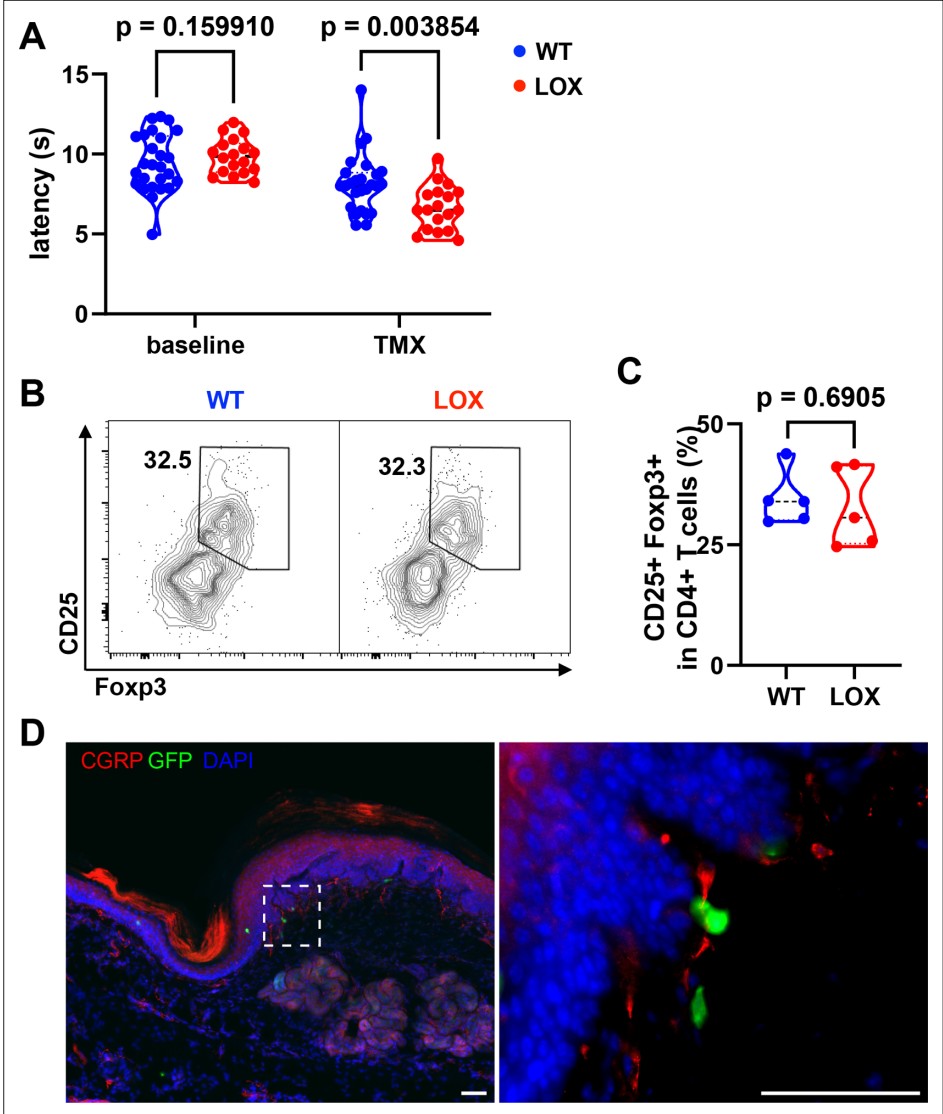

**Figure 5.** Heat hyperalgesia in mice deficient for *Penk* in regulatory T cell (Treg). (**A**) Withdrawal latency of wild-type (WT) and LOX mice before (baseline) and after administration of Tamoxifen (TMX). Each dot corresponds to the mean latency response (in seconds) of four measurements taken before TMX administration (baseline) and four measurements taken from day 7 onwards (TMX). Statistical modeling was performed using a non-parametric unpaired Mann–Whitney *t*-test with multiple corrections. The results shown in this figure are cumulative from two independent experiments with a total of 44 mice (26 WT and 18 LOX). Each dot is a mouse. (**B**) Representative flow cytometry contour plot of CD25 and Foxp3 staining on pad skin CD45$^+$CD3$^+$CD4$^+$ cells from WT or LOX mice 17 days after TMX gavage. (**C**) Quantification of the frequency of Treg (Foxp3$^+$CD25$^+$) among CD4$^+$ T cells in WT and LOX mice pad as shown in (**B**). The indicated p value was determined by a Mann–Whitney test. Each dot is a mouse from a single experiment. (**D**) Immunofluorescence staining of Calcitonin Gene-Related Peptide (CGRP) neurons (red), Foxp3-GFP cells (green), and 4'6-diamidino-2-phenylindole (DAPI) (blue) of footpad skin section of a female LOX mouse (scale bar represents 50 μm). The right panel is the magnification of the area indicated on the left panel.

The online version of this article includes the following figure supplement(s) for figure 5:

**Figure supplement 1.** Specific deletion of *Penk* in regulatory T cell (Treg) of LOX mice after Tamoxifen (TMX) administration.

**Figure supplement 2.** Heat hyperalgesia in wild-type (WT) and LOX mice.

as light touch and proprioception (sticky tape test), fail to show any significant effect of Treg-specific *Penk* deficiency (*Figure 5—figure supplement 2E–H*). Importantly, TMX administration does not alter the proportions of skin Treg in LOX compared to WT mice (*Figure 5B, C*), indicating that hyperalgesia does not result from an altered distribution of Treg.

Given this result, we next investigated whether Treg could be localized in contact with nociceptive neurons in the skin. Interestingly, some skin Tregs marked by the GFP reporter molecule in our WT and LOX mice can be observed in close contact with free nerve endings labeled with a Calcitonin Gene-Related Peptide (CGRP)-specific monoclonal antibodies (mAb) in the pad skin (*Figure 5D*). Because sensory neurons expressing CGRP are essential for noxious thermal heat, but not mechanical sensitivity (*McCoy et al., 2013*), this result suggests that Treg producing enkephalins could act locally on nociception.

## Discussion

The proenkephalin gene *Penk* encodes the precursor of opioid peptides with analgesic properties (*McLaughlin, 2013*). Enriched expression of *Penk* in Treg has been previously reported in several specific contexts, including TCR-transgenic mice (*Zelenika et al., 2002*), UVB exposure (*Shime et al., 2020*) or in the brains of mice recovering from stroke (*Ito et al., 2019*). We first explore the possible molecular mechanisms that may explain the preferential expression of *Penk* observed in tissue Treg. Using data mining and gene correlation analysis, we observed that TNFR and Batf might be involved in *Penk* regulation. Batf is known to regulate several genes through partnering with AP-1 and Irf-4 (*Murphy et al., 2013*), and we noted that several ChIP-Seq studies have reported the binding of these transcription factors in regulatory regions of *Penk*. Thus, our results support the hypothesis that TNFR signaling may regulate *Penk* expression in murine T cells through cooperation between Batf, AP-1 and/or Irf-4. Supporting this hypothesis, the AP-1 members Fos and Jun are crucial in *Penk* regulation in the murine hippocampus (*Sonnenberg et al., 1989*). Additionally, analysis of Penk-Cre reporter mice led us to conclude that *Penk* mRNA is predominantly found in tissue Treg, further supporting the hypothesis that Batf might be a chief regulator of *Penk*. Although *Penk* might be preferentially expressed by activated T cells at steady state (Treg and Tconv), its distribution may be broader in an inflammatory context. Consistent with this, it has been reported that IL-4-treated macrophages are able to reduce neuropathic pain through their ability to produce opioid peptides (*Celik et al., 2020*; *Pannell et al., 2016*).

A prior study attributed a function to UVB-exposed Treg-derived *Penk* in promoting the growth of epidermal keratinocytes in vitro and facilitating wound-healing in vivo (*Shime et al., 2020*). Consequently, heightened heat sensitivity in the absence of Treg-derived *Penk* may result from altered keratinocyte homeostasis in vivo. However, the impact of Treg-derived *Penk* on keratinocyte homeostasis in vivo under normal conditions has yet to be conclusively demonstrated.

Additionally, a similar hypersensitivity to heat has been recently reported in mice lacking *Penk* in Treg following TMX treatment (*Mendoza et al., 2024*) but not in another model of bone marrow chimeras, which allows the depletion of *Penk*-expressing Treg only in the dorsal root ganglion (DRG), a crucial relay of nociception. Instead, an increased mechanical allodynia was observed in that model (*Midavaine et al., 2024*), a result that we did not observe. This discrepancy might be explained by the relatively low number of animals that we tested for mechanical allodynia. It is also possible that the deletion of Penk with TMX by oral gavage might have been less efficient in the DRG than in lymphoid organs.

Nevertheless, Treg-derived enkephalins might regulate pain at multiple sites, including the DRG and peripheral tissues, such as the skin or the colon. Related to the colon, Penk expression by CD4[+] T cells has been linked to analgesia in murine models of visceral pain (*Basso et al., 2018*; *Basso et al., 2016*; *Boué et al., 2014*). However, the specific involvement of Treg in this process has yet to be investigated.

Furthermore, the hypothesis that skin Treg directly modulates pain is supported by our observations that skin Treg (1) expressed the highest level of *Penk* and (2) are observed in contact with CGRP-expressing sensory neurons. Notably, *Penk*-expressing Tregs are closer to neurons in the skin than non-expressing Tregs (*Mendoza et al., 2024*). Experiments are underway to formally demonstrate that enkephalins produced by Treg directly impact the electrophysiological potential of isolated neurons.

In neuroinflammatory settings, such as sciatic nerve constriction, the pain modulation by Treg could be contingent upon their immunosuppressive function (*Davoli-Ferreira et al., 2020*; *Duffy et al., 2019*; *Ledeboer et al., 2007*; *Shen et al., 2013*), potentially mediated through the secretion of IL-10 or IL-35 (*Davoli-Ferreira et al., 2020*; *Duffy et al., 2019*). Contrary to this notion, mice lacking IL-10 in Tregs do not suffer from heat hyperalgesia (*Mendoza et al., 2024*), indicating that the control of inflammation by IL-10 is not responsible for pain modulation by Treg. Interestingly, IL-10 also possesses analgesic properties in murine models of cisplatin-induced neuropathic pain (*Laumet et al., 2020*). The observation that mice with IL-10-KO Tregs experienced similar pain levels to controls further indicates that IL-10 produced by Treg cannot compensate for the absence of *Penk*, highlighting the unique role of Treg-derived enkephalins on nociception.

Finally, our in vitro and in vivo experiments, along with results from *Midavaine et al., 2024*, show that Tregs maintains their suppressive capacities in the absence of *Penk*, ruling out the possibility that hyperalgesia stems from increased inflammation due to a defect in the immunosuppressive function of Treg. Instead, our results strongly indicate a direct implication of enkephalins produced by Treg in nociception, revealing a novel non-immune intrinsic role of Treg in the endogenous regulation of basal somatic sensitivity.

## Materials and methods
### Extraction of Treg meta-signatures
The datasets used were selected based on a 'Treg* AND (Tconv* OR Teff*) AND *Mus musculus*' search in the GEO dataset website (https://www.ncbi.nlm.nih.gov/gds). GEO datasets were manually inspected for inclusion of studies comparing fresh Treg with fresh Tconv from lymphoid organs. Characteristics of selected datasets are summarized in *Table 1*. For each dataset, we generated a list of differentially expressed genes with a cutoff based on a false discovery rate inferior to 0.05 and a log2 fold change superior to 1 with the GEO2R embedded algorithm.

### Mice
All male and female mice were on a C57Bl/6J background. *Foxp3*^tm9(EGFP/cre/ERT2)Ayr/J (*Foxp3*^Cre-ERT2) (catalog #016961), *B6.Cg-Gt(ROSA)26Sor*^tm9(CAG-tdTomato)Hze/J (*ROSA*^tdTomato) (catalog #007909), and *B6. Cg-Penk*^tm1.1(cre/ERT2)Hze/J (*Penk*^Cre-ERT2) (catalog #022862) were purchased from The Jackson Laboratory. *C57BL/6JSmoc-Penk*^em1(flox)Smoc (*Penk*^flox) were purchased from Shanghai Model Organisms (catalog NM-CKO-210032). Bone marrow from Penk-KO and WT littermates mice was a kind gift of Dr G Dietrich (IRSD, Toulouse, France). *Penk*-deficient bone marrow cells were purified from the B6.129-Penk-rs^tm1Pig/J strain (The Jackson Laboratory, catalog #002880). For the *Penk* mapping experiments, *Penk*^Cre-ERT2 were bred with *ROSA*^tdTomato. All mice were confirmed to be homozygous for the inducible Cre and at least heterozygous for tdTomato by touchdown Polymerase Chain Reaction (primers sequences available on request). To specifically knock-out the *Penk* gene in Tregs, *Foxp3*^Cre-ERT2 were crossed with *Penk*^flox leading to double heterozygous mice (F1) that were crossed together resulting in double homozygous F2 littermates. All mice used in this study were of F3 generation. Mice were genotyped by touchdown PCR (primers sequences available on request). TMX (Thermo Fisher, Les Ulis, France) was dissolved in peanut oil at the concentration of 40 mg/ml under 37°C agitation and delivered by 200 µl oral gavage. All mice were administered TMX only once. Mice were housed under specific pathogen-free conditions and were used for experiments at 8 weeks or older. Mice were exposed to a 12-hr light and 12-hr dark cycle. Protocols are approved by the Ethics Committee for Animal Experimentation Charles Darwin (APAFIS #32284-2021070513305185 v5) and by the Ethics Committee 002 (#11837-2017101816028463 v5).

### Preparation of cell suspensions
Organs were harvested on ice in phosphate-buffered saline (PBS) 3% fetal calf serum (FCS). Inguinal, brachial, and mesenteric LNs and spleens were directly mashed through a 70-µm filter and suspended in PBS 3% FCS. Lungs, colons, livers, visceral adipose tissue, and skin were dissected, minced then incubated in appropriate digestion buffers (Miltenyi Biotec, Paris, France) at 37°C for various duration according to manufacturer protocols. Cell suspensions were then passed through a 70-µm cell strainer and suspended in PBS 3% FCS. To eliminate dead cells and debris, liver cell suspensions were isolated

on a 70:30 Percoll gradient. Rings were collected, washed, and cell pellets were suspended in PBS 3% FCS. ACK Lysing Buffer was used to eliminate red blood cells in the lungs, livers, and spleens (lysis performed for 1 min at room temperature [RT], followed by two washes with PBS), prior to staining for flow cytometry.

## Antibodies and flow cytometry analysis

The mAb and fluorescent reagents used in this study are listed in *Supplementary file 1*. Up to $4 \times 10^6$ cells were incubated for 30 min at 4°C with fixable live/dead dye and with the anti-CD16/CD32 (clone 2.4G2) to block FcgRIII and FcgRII receptors, respectively. Cells were then stained with a combination of antibodies. Cell-surface staining was performed in PBS 3% FCS for 20 min at 4°C. Permeabilization and intracellular staining were performed using the Foxp3/Transcription Factor Staining Buffer Set kit and protocol (Thermo Fisher #00-5521-00, Les Ulis, France). Stained cells were washed with PBS 1× before acquisition on a Cytek Aurora flow cytometer (Cytek Bioscience, Fremont, CA, USA). UMAPs were generated using FlowJo software, version 10.8.1 (TreeStar, Ashland, OR, USA).

## Flow-Fluorescent In Situ Hybridization (FISH) staining

Frozen splenocytes were thawed and washed once in complete RPMI (Thermo Fisher) supplemented with 10% vol/vol FCS (Thermo Fisher). Flow-FISH staining was performed using the PrimeFlow RNA Assay Kit (Thermo Fisher #88-18005-210) and the subsequent anti-mouse Penk probes set (Thermo Fisher #PF-204) according to the manufacturer's protocol. For flow cytometry staining, after Fc receptor saturation (anti-mouse CD16/32, clone 93, 1:50, Biolegend) and dead cells detection (Fixable Viability Dye, 1:1000, Thermo Fisher), cells were surface labeled for 30 min on ice with the following antibodies: CD8a-BV421 (Clone 53-6.7, 1:200, BD Biosciences #563898), CD44-BV605 (Clone IM7, 1:200, BD Biosciences #563058), CD3e-BV711 (Clone 145-2C11, 1:50, Biolegend #100349), CD4-PE-CF594 (Clone RM4-5, 1:800, BD Biosciences #562285), and CD62L-PE-Cy7 (Clone MEL-14, 1:300, BD Biosciences #560516). Intracellular staining with Foxp3-PE antibody (Clone NRRF-30, 1:100, Thermo Fisher #12-4771-82) was performed after fixation and permeabilization with the reagents provided in the kit. A second fixation was performed with the reagents provided in the kit before proceeding to the FISH staining. For FISH staining, cells were incubated with the anti-mouse Penk probes set (1:20) for 2 hr at 40°C. After washes, cells were kept overnight at 4°C in a wash buffer containing RNase inhibitors. The day after, amplification steps were performed to increase the signal: cells were first incubated with pre-amplification mix during 1.5 hr at 40°C, washed and then incubated with amplification mix for an additional 1.5 hr at 40°C. Cells were then incubated with the label probes (Alexa Fluor 647, 1:100) for 1 hr at 40°C. Data were acquired on a 4-lasers LSR-Fortessa (BD Biosciences) and fcs files were analyzed with FlowJo as above.

## In vitro and in vivo suppression assay

Cells from the spleen and the LNs of Foxp3-GFP mice were isolated and enriched for T cells with LS column (Miltenyi Biotec) using anti-CD19-biotin (Clone 1D3, BD Biosciences #553784), aCD11b-biotin (Clone M1/70, BD Biosciences #553309), and anti-Ter119-biotin (Clone TER-119, BD Biosciences #553672). Afterward, enriched T cells were stained with anti-CD4 PE (Clone RM4-5, 1:400, BD Biosciences #553048) and anti-CD62L AF700 (Clone MEL-14, 1:100, BD Biosciences #560517) and sorted as CD62L⁺CD4⁺GFP⁺ (Treg) or GFP⁻ (Tconv) fractions on a FACS Aria III (BD Biosciences, France). Then, Tconv and Treg were labeled with CellTrace Violet and CFSE (Thermo Fisher #C34557 and C34554), respectively, before the culture. For T-cell activation, either CD3/CD28 DynaBeads (Thermo Fisher #11452D) or soluble anti-CD3 (1 µg/ml) + splenocytes from Rag2$^{-/-}$ mice were used. When specified, Naloxone hydrochloride (Tocris, #0599/100) was added to the media at the indicated concentration at the beginning of the culture. At the end of the culture, cells were stained with eF780 Fixable Viability Dye (Thermo Fisher #65-0865-14) before acquisition on a LSR Fortessa (BD Biosciences). To induce colitis, C57Bl/6 Rag2$^{-/-}$ mice were intraperitoneally injected with CD4⁺CD62L$^{hi}$GFP⁻ naive T cells ($4 \times 10^5$ cells per mouse) isolated from spleens and LNs of WT C57Bl/6 Foxp3-GFP mice (kind gift of Dr B. Salomon), together with or without CD4⁺CD25$^{hi}$CD62L$^{hi}$ Treg cells ($2 \times 10^5$ cells per mouse) isolated from spleens of Rag2$^{-/-}$ mice reconstituted with bone marrow cells of Penk-WT or Penk-KO mice.

## Immunofluorescence

Mice were anesthetized with a mix of 100 mg/kg ketamine and 20 mg/kg xylazine, decapitated with scissors and the glabrous skin of the hindpaw harvested. The skin was then fixed with paraformaldehyde 4% for 2 hr at RT. Fixed tissue was cryoprotected in 30% sucrose containing 0.05% sodium azide diluted in PBS at 4°C for 48 hr, and cut with a cryostat (Microm HM550) into 25 µm sections placed directly onto gelatine coated slides. For fluorescent immunostaining, slides were washed with PBS and blocked with 1% bovine serum albumin in PBS + 0.2% Triton-X (PBS-T) for 1 hr at RT, then incubated with primary antibodies diluted in PBS-T overnight at 4°C. Sections were then washed in PBS and incubated with secondary antibodies diluted 1:1000 in PBS-T, for 2 hr at RT. Slices were finally washed in PBS and cover-slipped with Fluoromount-G containing DAPI (Invitrogen 00-4959-52). The following primary antibodies were used for immunofluorescence staining at the following dilutions: anti-GFP raised in rabbit (1:1000, Chromotek # PABG-1), anti-CGRP raised in goat (1:500, Bio-Rad #1720-9007). We also used the following secondary antibodies: anti-rabbit conjugated with Cy3 raised in donkey (Jackson ImmunoResearch, #711-165-152), anti-rabbit conjugated with Alexa-488 raised in donkey (Jackson ImmunoResearch, #711-545-152), and anti-goat conjugated with Cy5 raised in donkey (Jackson ImmunoResearch, #705-175-147). Images were obtained by epifluorescent microscopy with a motorized fluorescence microscope Axio Imager M2 equipped with a camera Axiocam 705 (Zeiss). Skin slices were imaged using a Colibri 7 light source, ×10/0.30 and ×63/1.25 objectives and the following filters from Zeiss: 02 DAPI, 38 HE eGFP, 43 HE DsRed, and 50 Cy5 BP640/30. Images were generated with Zen blue 3.4 (Zeiss) and brightness and contrast adjusted using ImageJ/Fiji.

## Behavioral tests

The methods used were described by *Baker et al., 2002* and *Peirs et al., 2021*.

### Heat sensitivity

For the hot plate test, the BIO-CHP apparatus was used (BIOSEB, France). Mice were placed on a metal plate maintained at 55°C. The response latency, which is the time taken to observe a nocifensive behavior such as jumping, licking, or flicking of the hind paws, was recorded. The mice were then immediately removed from the plate upon the recording of a reaction, or within 25 s if no response was observed to prevent tissue damage. The test was repeated every 2 days for a total of four measures before the administration of TMX and four measures starting from D3 post-TMX. Mice remained in their home cage except when being tested on the hot plate. The experimenter was blind to the genotype of the mice until the end of the experiment. Noxious heat sensitivity was independently assessed in the Neurodol laboratory with an Hargreaves test. Animals were placed in an acrylic chamber on a heated (30°C) glass plate and acclimated to the test chamber for 30 min during 2 days and then for 30 min on the third day prior to testing. Using a plantar analgesia meter (IITC, 40% intensity), a radiant heat source of constant intensity was focused on the plantar surface of the hind paw and the latency of paw withdrawal measured. The heat source was stopped upon paw withdrawal with a cutoff of 20 s to avoid injury. Heat sensitivity test was repeated three times on each hind paw with a 5-min interval between tests and the results for each paw were averaged together.

### Static mechanical sensitivity

Static mechanical sensitivity was assessed with a Von Frey test. Briefly, mice were habituated to transparent Plexiglas chambers on an elevated wire mesh table for at least 30 min for 2 days, and prior to testing. Assessments were performed using a set of calibrated Von Frey monofilaments using the Up-Down method, starting with the 0.4 g filament. Each filament was gently applied to the plantar surface of the hind paw for 3 s or until a response such as a sharp withdrawal, shaking, or licking of the limb was observed. Rearing or normal ambulation during filament application were not counted. Filaments were applied at 5-min intervals until the threshold was determined. The 50% paw withdrawal threshold (PWT) was determined for each mouse on both hind paws.

### Dynamic mechanical sensitivity

Dynamic mechanical sensitivity was assessed with the cotton swab test. Briefly, animals were habituated to transparent Plexiglas chambers on an elevated wire mesh table as described above. The head

of a cotton swab was teased and puffed out with forceps until it reached approximately three times its original size. Tests were performed by lightly moving the cotton swab across the surface of the hind paw from heel to toe. If the animal reacted (lifting, shaking, licking of the paw) a positive response was recorded. A negative response was recorded if the animal showed no such behavior. The application was repeated six times with a 5-min interval between applications, and a percentage of positive responses was determined. Paw withdrawal frequency (PWF) was determined for both hind paws of each mouse.

### Noxious mechanical sensitivity

Noxious mechanical sensitivity was assessed with the pinprick test. Animals were acclimated to transparent Plexiglas chambers placed on a wire mesh table as described above. A small insect pin (tip diameter = 0.03 mm) was applied 10 times with a 5-min interval between applications on the plantar side of each hind paw. Care was taken to apply a minimal pressure without penetrating the skin. If the animal showed aversive behavior (lifting, shaking, licking of the paw) a positive response was recorded. A negative response was recorded if the animal showed none of these reactions within 1 s of application and a percentage of positive responses were determined. PWF was calculated by averaging the positive responses for each mouse for each hind paw.

### Proprioception test

Animals were placed in an empty plastic cage and allowed to acclimate for 15–20 min. A 8-mm diameter adhesive paper circle was then applied to the plantar surface of the hind paw covering the footpads, and the animals were immediately placed back in the chamber. The animals were observed until they demonstrated a behavioral response to the adhesive tape, and the latency in seconds to respond was recorded. Inspection of the paw, shaking of the paw or attempting to remove the tape were all considered valid responses. Each animal was habituated one time the day prior testing, and then tested three times with a 5-min interval between tests, and the three values averaged for each animal for each hind paw.

## Statistical analysis

All statistical tests are reported in the figure legends and were performed with Prism v9.4.1 (GraphPad Inc, La Jolla, CA, USA). The statistical power of the analyses (alpha) was set at 0.05. No a priori sample size estimation based on beta power (1-alpha) was performed.

## Acknowledgements

The authors would like to express their deepest gratitude to Dr Stéphane Melik-Parsadaniantz for the kind gift of essential tools used in this study and to Pr Radhouane Dallel (NeuroDol, UCA) for his help in setting up the collaboration. We would like to thank Olivier Brégerie, Flora Issert, Doriane Foret, Claire Lacoste (UMS 28, Paris), and Sylviane Rousselin (Neurodol, Clermont-Fd) for taking care of our mice and for their support filling ethical applications; Dr Morgane Hilaire (CIMI-PARIS) and Amélie Descheemaeker (NeuroDol) for technical help, Anne-Marie Gaydier for secretarial assistance, Francois-Xavier Lejeune (ICM, Paris) for help with biostatistics analysis, Dr Jean-Luc Teillaud for critical review of the manuscript, and Dr Benoit Salomon for providing Foxp3$^{Cre-ERT2}$, ROSA$^{tdTomato}$, and Foxp3$^{GFP}$ mice and for many years of enjoyable collaborations and discussions. This study was funded by research grants from Sorbonne University (SU) (Emergence 2021), la Ligue Nationale contre le Cancer (LNC) to GM, and by the National Institute for Health and Medical Research (INSERM). NA was supported by a doctoral fellowship from Fondation ARC and by a postdoctoral fellowship from EGLE TX, MF by a doctoral fellowship from the French Ministère de l'Enseignement Supérieur et de la Recherche, AA by a doctoral fellowship from the French Ministère de l'Enseignement Supérieur et de la Recherche and a 4th year extension from Fondation pour la Recherche Médicale (FRM), MP and MN by pre graduates fellowships from SU and EGLE TX, respectively. MN is supported by a doctoral fellowship from the IUC of SU. JN is supported by a postdoctoral salary from the LNC. The funders had no role in the supervision of the research, in the analysis, or interpretation of the results.

# Additional information

## Funding

| Funder | Grant reference number | Author |
|---|---|---|
| Ligue nationale contre le cancer | | Gilles Marodon |
| Sorbonne Université | Emergence | Gilles Marodon |
| INSERM | | Gilles Marodon |

The funders had no role in study design, data collection, and interpretation, or the decision to submit the work for publication.

## Author contributions

Nicolas Aubert, Conceptualization, Data curation, Formal analysis, Validation, Investigation, Visualization, Methodology, Writing - original draft, Writing - review and editing; Madeleine Purcarea, Data curation, Formal analysis, Investigation, Methodology; Julien Novarino, Conceptualization, Data curation, Formal analysis, Supervision, Validation, Investigation, Visualization, Methodology, Writing - review and editing; Julien Schopp, Alexis Audibert, Formal analysis, Investigation, Visualization, Methodology; Wangtianrui Li, Data curation, Formal analysis, Investigation, Visualization, Methodology; Marie Fornier, Léonie Cagnet, Marie Naturel, Armanda Casrouge, Investigation, Methodology; Marie-Caroline Dieu-Nosjean, Writing - review and editing; Nicolas Blanchard, Investigation, Methodology, Writing - review and editing; Gilles Dietrich, Resources, Validation, Writing - review and editing; Cedric Peirs, Resources, Supervision, Visualization, Writing - review and editing; Gilles Marodon, Conceptualization, Formal analysis, Supervision, Funding acquisition, Investigation, Writing - original draft, Project administration, Writing - review and editing

## Author ORCIDs

Alexis Audibert ⓘ https://orcid.org/0009-0006-5099-958X
Gilles Dietrich ⓘ https://orcid.org/0000-0002-2232-1716
Cedric Peirs ⓘ https://orcid.org/0000-0002-2296-0323
Gilles Marodon ⓘ https://orcid.org/0000-0003-4889-6785

## Ethics

Protocols are approved by the Ethics Committee for Animal Experimentation Charles Darwin (APAFIS #32284-2021070513305185 v5) and by the Ethics Committee 002 (#11837-2017101816028463 v5).

Public Review: https://doi.org/10.7554/eLife.91359.3.sa1
Author response https://doi.org/10.7554/eLife.91359.3.sa2

# Additional files

## Supplementary files

• Supplementary file 1. Monoclonal antibodies used in the study of Penk$^{Cre}$ × ROSA26$^{tdTomato}$. Indicated are the specificity of the antibody with the fluorescent dye, the provider, the catalog number, the clone, and the dilution of the monoclonal antibodies (mAb) used to generate results of *Figure 3*.

• MDAR checklist

• Source data 1. List of the differentially expressed genes between conventional T cell (Tconv) and regulatory T cell (Treg). Only genes upregulated in 10 out of 11 datasets are presented with the associated log2 fold change as determined by GEO2R (NA = not available).

## Data availability

The dataset used in the publication to establish the meta Treg signature (Figure 1A) is available in Source data 1. GSE accession numbers are provided in Table 1 or Figure legends when applicable.

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
