## [Editor Report · eLife assessment]

This study presents a **valuable** finding on a new role of Foxp3+ regulatory T cells in sensory perception, which may have an impact on our understanding of somatosensory perception. The authors identified a previously unappreciated action of enkephalins released by immune cells in the resolution of pain and several upstream signals that can regulate the expression of the proenkephalin gene PENK in Foxp3+ Tregs. The generation of transgenic mice with conditional deletion of PENK in Foxp3+ cells and PENK fate-mapping is novel and generates **compelling** data; they also show a comprehensive analysis of Tregs in control and transgenic mice, longitudinal data on heat sensitivity and co-localization of PENK+ Tregs with thermal sensory neurons in the skin further supporting their hypothesis. The study would be of interest to the biologists working in the field of neuroimmunology and inflammation.

---

## [Referee Report · Public Review]

The study addresses the role of enkephalins, which are specifically expressed by regulatory T cells (Treg), in sensory perception in mice. The authors used a combination of transcriptomic databases available online to characterize the molecular signature of Treg. The proenkephalin gene Penk is among the most enriched transcripts, suggesting that Treg plays an analgesic role through the release of endogenous opioids. In addition, in silico analysis suggests that Penk is regulated by the TNFR superfamily; this being experimentally confirmed. Using flow cytometry analysis, the authors then show that Penk is mostly expressed in Treg of the skin and colon, compared to other immune cells. Finally, genetic conditional excision of Penk, selectively in Treg, results in heat hypersensitivity, as assessed by behavior analysis.

Editors' note: The authors accepted most if not all the suggestions given by the reviewers and the revised version of the manuscript is substantially improved.

---

## [Author Response]

The following is the authors’ response to the original reviews.

**Public Reviews:**

**Reviewer #1 (Public Review):**
Summary:The authors explore mechanisms through which T-regs attenuate acute pain using a heat sensitivity paradigm. Analysis of available transcriptomic data revealed expression on the proenkephalin (Penk) gene in T-regs. The authors explore the contribution of T-reg Penk in the resolution of heat sensitivity.Strengths:Investigating the potential role of T-reg Penk in the resolution of acute pain is a strength.Weaknesses:The overall experimental design is superficial and lacks sufficient rigor to draw any meaningful conclusions.

We hope that the reviewer will reconsider this severe criticism after examining the updated manuscript and results.

For instance:(1) The were no TAM controls. What is the evidence that TAM does not alter heat-sensitive receptors.

the impact of TMX on heat perception is not the object of this study. Nevertheless, it appears that heat-sensitivity in controls WT (blue dots) is slightly diminished after TMX administration (Figure 5A), suggesting that heat-sensitive receptors are moderately altered by TMX per se. This reduction is much more pronounced for LOX mice. Thus, although it is possible that TMX play a marginal role on heat sensitivity by itself, the results show a much more pronounced effect of TMX in LOX than in WT, in favor of a role for Penk Treg in heat sensitivity.

(2) There are no controls demonstrating that recombination actually occurred. How do the authors know a single dose of TAM is sufficient?

these results are now presented in figure S4. A 70% reduction in Penk mRNA is observed in Treg after a single administration of TMX.

(3) Why was only heat sensitivity assessed? The behavioral tests are inadequate to derive any meaningful conclusions. Further, why wasn't the behavioral data plotted longitudinally

The longitudinal data are presented in figure S5A. New behavioral tests have been performed and the results are now shown in figure S5E-H. Importantly, heat sensitivity was observed in two independent laboratory with two different tests.

**Reviewer #2 (Public Review):**
Summary:The present study addresses the role of enkephalins, which are specifically expressed by regulatory T cells (Treg), in sensory perception in mice. The authors used a combination of transcriptomic databases available online to characterize the molecular signature of Treg. The proenkephalin gene Penk is among the most enriched transcripts, suggesting that Treg plays an analgesic role through the release of endogenous opioids. In addition, in silico analysis suggests that Penk is regulated by the TNFR superfamily; this being experimentally confirmed. Using flow cytometry analysis, the authors then show that Penk is mostly expressed in Treg of the skin and colon, compared to other immune cells. Finally, genetic conditional excision of Penk, selectively in Treg, results in heat hypersensitivity, as assessed by behavior analysis.Strengths:The manuscript is clear and reveals a previously unappreciated role of enkephalins, as released by immune cells, in sensory perception. The rationale in this manuscript is easy to follow, and conclusions are well supported by data.Weaknesses:The sensory deficit of Penk cKO appears to be quite limited compared to control littermates.
**Reviewer #3 (Public Review):**
Summary:Aubert et al investigated the role of PENK in regulatory T cells. Through the mining of publicly available transcriptome data, the authors confirmed that PENK expression is selectively enriched in regulatory but not conventional T cells. Further data mining suggested that OX40, 4-1BB as well as BATF, can regulate PENK expression in Tregs. The authors generated fate-mapping mice to confirm selective PENK expression in Tregs and activated effector T cells in the colon and spleen. Interestingly, transgenic mice with conditional deletion of PENK in Tregs resulted in hypersensitivity to heat, which the authors attributed to heat hyperalgesia.Strengths:The generation of transgenic mice with conditional deletion of PENK in foxp3 and PENK fate-mapping is novel and can potentially yield significant findings. The identification of upstream signals that regulate PENK is interesting but unlikely to be the main reason why PENK is predominantly expressed in Tregs as both BATF and TNFR are expressed in effector T cells.Weaknesses:There is a lack of direct evidence and detailed analysis of Tregs in the control and transgenic mice to support the authors' hypothesis. PENK was previously reported to be expressed in skin Tregs and play a significant role in regulating skin homeostasis: this should be considered as an alternative mechanism that may explain the changed sensitivity to heat observed in the paper.

We now provide a detailed analysis of Treg with or without Penk, from their immunosuppressive functions to their colocalization with sensory neurons in the skin, supporting their function as natural analgesics. The alternate hypothesis relative to skin homeostasis is now clearly presented and discussed.

**Recommendations for the authors:**

**Reviewer #2 (Recommendations For The Authors):**
Most of my comments should be addressable in a revised manuscript but will require additional analysis.Major:- According to flow cytometry analysis, Penk is expressed mostly in Treg of the skin and colon. What may account for such restricted expression? Where could Treg-released enkephalins act?

We now rephrased the paper to emphasize the known role of Batf in tissue Treg differentiation. We believe the Batf dependency of Penk expression is the reason why tissue Treg are more enriched in Penk than Treg from lymphoid organs. This is now clearly discussed.

We also provide a new figure (Figure S1) that shows that binding of Batf and co factors AP1 and IRF4 were reported to bind to Penk regulatory regions. Altogether, the role of Batf in tissue Treg differentiation would explain why tissue Treg such as colon and skin are particularly enriched in Penk. This is now clearly stated in the revised manuscript.

As to know where Treg-released enkephalins act, we performed immunostainings in the skin and observed that Treg could colocalize with sensory neurons (shown in a new figure 5, panel D). This observation raise the hypothesis that Treg-released enkephalins could act on sensory neurons locally.

- Which mechanism can underlie heat hypersensitivity in Penk cKO mice? Which sensory neurons are involved? Are other sensory modalities affected, such as mechanical sensitivity?

As stated above, we show that Treg can be in close contact with thermal sensors neurons producing CGRP. These data are shown in figure 5D. We have also tested may other nociceptive stimulus (innocuous and noxious) and did not detect significant differences. These data are presented as a supplementary figure S5. Whether enkephalins produced by Treg can change the stimulation threshold of various nervous fibers is currently performed by electrophysiology.

- No control is provided to ensure that Penk is selectively excised in Treg cells in cKO mice.

We have performed additional experiments with fluorescent probes to document Penk mRNA expression in cKO mice. The results on the specific expression of Penk mRNA in various subsets post-TMX are shown in a supplementary figure S4.

- The authors acknowledge that Penk from Treg was previously studied in an animal model of inflammatory pain. However, which role these endogenous opioids play is unclear, especially since authors discovered that enkephalins are likely continuously released at steady states. This is not enough discussed in the narrative, which surprisingly does not separate the results from the discussion.

The results and discussion are now separated in two sections.

Minors:- Replace "Fox3 1" with "Fox31" (line 31), "functions 15" with "functions15" (line 43), "BATF 19" with "BATF19" (line 85).- Text mentions Figure S4 (line 125), which is most likely S3.
**Reviewer #3 (Recommendations For The Authors):**
Given the most significant finding of this paper is based on the heat-induced pain model, there is surprisingly little analysis of Tregs in this context. The authors analyzed spleen and colon Tregs at steady state, it is unclear whether any of these Tregs are involved in pain sensitivity directly. Skin Tregs or other relevant Tregs to this model should be analyzed in control and Lox mice. This is particularly relevant as PENK expression was previously reported in skin Tregs and plays a significant role in skin homeostasis (Yamazaki et al 2020 PNAS). Does PENK conditional deletion alter Treg frequencies, numbers, and immune suppressive function? Not even spleen or colon Treg were analyzed comparing control and lox mice.

We now provide evidences showing unaltered immunosuppressive functions of Treg in the absence of Penk (Figure 4), and more importantly unaffected proportions of skin Treg in mice lacking Penk in Treg, at the very site of heat stimulation (Figure 5B-C). We also observed unaffected representation of Treg in the spleen and lymph nodes, but we do not feel that these data are necessary to interpret the results.

Given the role of PENK in skin Tregs, could the observed effect in Figure 4 be due to altered skin homeostasis rather than sensitivity to pain?

The reviewer is referring to a paper where Penk in skin Treg play a role on UV-damaged keratinocytes in vivo (Shime et al., 2020, PNAS). To our knowledge, a role for Penk produced by skin Treg on keratinocytes homeostasis at the steady state is currently unknown. Nevertheless, this hypothesis is now clearly stated and discussed in the manuscript.

The authors stated that only after 7 days post tamoxifen treatment was heat hyperalgesia observed: deletion of PENK in Treg but not Tconv should be confirmed: is deletion only complete after 7 days or is the effect observed due to indirect effects of altered "normal" Treg function?

We have performed a kinetics to document Penk deletion at D3, D7 and 30 post-TMX. Results show a specific deletion of Penk in Treg at all time points so we combined all the time points for the representation of the results (Figure S4). As for the indirect effects of “altered” normal function, we now provide the reader with a new figure (Figure 4), showing that Penk deficient Treg are not impaired in their suppressive function in vitro and in vivo.

Actual protein/peptide production of enkephalins by Tregs should be confirmed. It is also unclear which peptide(s) can be secreted and presumably responsible for the changes in heat sensitivity.

This is a very interesting question that we addressed with a MENK ELISA but without success at reproducing the results. An ongoing project will use mass spectrometry to fully characterize the peptides produced by Treg and activated Tconv.

The analysis of PENK regulation by Tregs is interesting despite them being entirely based on data mining. BATF is a pioneering factor expressed by all activated effector T cells. While the connection between BATF and PENK may explain why the authors observed PENK expression chiefly in activated effectors and Tregs, BATF cannot be the reason why PENK is "predominantly" expressed by Tregs. Similarly, 4-1BB and OX40 can be induced on effector T cells. Is PENK under the control of Foxp3? There are lots of publically available datasets on Foxp3/IL-2 dependent Treg signatures through which this can be addressed.

We now provide a supplementary figure (Figure S1), showing a compilation of ChIP Seq studies for various transcription factors in various T cell subsets. We provide the reader with a list of all the TF that have been reported to bind in the regulatory regions of Penk. In agreement with our hypothesis, BATF, FOXP3, IRF4 and several others are present in that list. Further work is needed to decipher the exact contribution of each of those TF to the regulation of Penk in Treg vs activated Tconv that is beyond the scope of this report.